# Tolerability, Safety and Efficacy of a Specific Rehabilitation Treatment Protocol for Axillary Web Syndrome: An Observational Retrospective Study

**DOI:** 10.3390/cancers15020426

**Published:** 2023-01-09

**Authors:** Margherita Beatrice Borg, Laura Mittino, Marco Battaglia, Alberto Loro, Laura Lanzotti, Marco Invernizzi, Alessio Baricich

**Affiliations:** 1Physical and Rehabilitation Medicine, Department of Health Sciences, University of Eastern Piedmont, 28100 Novara, Italy; 2Physical and Rehabiliation Medicine Division, Azienda Ospedaliero-Universitaria “Maggiore della Carità”, 28100 Novara, Italy; 3Dipartimento Attività Integrate Ricerca e Innovazione (DAIRI), Translational Medicine, Azienda Ospedaliera SS. Antonio e Biagio e Cesare Arrigo, 15121 Alessandria, Italy

**Keywords:** axillary web syndrome, breast cancer, rehabilitation, pain, physical therapy, manual lymphatic drainage

## Abstract

**Simple Summary:**

Axillary web syndrome (AWS) is a common post-surgical sequela of breast cancer (BC) treatment. It determines pain and it limits the upper limb range of motion (ROM), reducing the quality of life of BC survivors. To date, various treatments and combinations of rehabilitative approaches have been proposed, but standardized guidelines are still lacking. The aim of our retrospective study was to assess the safety and tolerability of our specific AWS rehabilitative treatment protocol, and to determine its efficacy in reducing pain and improving upper limb ROM. We found that 60-min rehabilitative sessions composed of manual lymphatic drainage, stretching and mobility exercises, and soft tissue mobilization, performed three times/week until clinical resolution, represents a safe and well-tolerated rehabilitation protocol. Moreover, 98% of the patients reached complete shoulder ROM in flexion and abduction after the treatment, and there was a significant reduction in pain.

**Abstract:**

Axillary web syndrome (AWS) is a highly prevalent surgical complication affecting BC survivors. It presents as a subcutaneous cording that limits the upper limb range of motion (ROM) and causes pain. Its etiology is still debated, and its treatment is not well defined. Therefore, we aimed to investigate the safety, tolerability and efficacy of our specific AWS rehabilitative treatment protocol. We conducted an observational retrospective study on a cohort of 92 AWS patients referred to the oncological outpatient service of a university hospital. We collected data from medical records before (T0) and after (T1) the treatment. The studied protocol was composed of 60-min sessions, carried out 3 times/week by specialized physiotherapists, until the clinical resolution of AWS. We found that a mean of 8.74 ± 2.12 rehabilitative sessions were needed, and only one patient stopped early. At T1, shoulder ROM was complete in both abduction and flexion in 98% of patients; AWS was no longer detectable in 64% of them, and pain significantly decreased compared to T0. In conclusion, our protocol proved to be safe, well-tolerated and seemed to be effective in treating AWS.

## 1. Introduction

Breast cancer (BC) is the most common female tumor, with an estimated 2.3 million new cases in 2020, and it represents the leading cause of death among young women [1,2,3]. In the last few decades, BC treatments have significantly improved, leading to a remarkable increase in its prognosis. Nowadays, 5-year survival rates are in the range of 90%, and 10-year survival is about 80% [4]. Thus, the clinical management of BC sequelae currently plays a pivotal role in preserving the wellbeing and health-related quality of life (HR-QoL) of BC survivors [5].

Among the long-term toxicity of BC treatment, cardiotoxicity is one of the most important, as it might occur years after the administration of adjuvant therapies, particularly anthracyclines and trastuzumab, and it might be life-threatening. Another important sequela to be kept in mind is the estrogen deficiency due to aromatase-inhibitor drugs in breast cancer patients with endocrine sensitive disease; bone, being an estrogen-dependent tissue, is strongly affected by estrogen circulating levels, and thus, decreasing levels can produce a rapid increase in the potential risk of fractures.

Among the mid-term side effects that have a strong impact on quality of life, fertility impairment is notably one of the main issues to be discussed and managed in young patients who are eligible for systemic adjuvant therapies. Therefore, fertility preservation techniques should be discussed with all young women who require adjuvant chemotherapy. Psychosocial changes in affected women are also widely reported, as the impact of a breast cancer diagnosis could induce depression, anxiety and intrusive thoughts [4].

Finally, considering short-term side effects of BC treatments, axillary web syndrome (AWS) is an important post-surgical sequela; another lymphatic vessel alterations-based post-surgical sequela is breast cancer-related lymphoedema (BCRL), which by contrast could also develop years or even decades after the surgery.

Axillary web syndrome (AWS) was first described in 2001 by Moskovitz et al. as a post-surgical sequala that affects BC patients who underwent axillary surgery. AWS was depicted as the presence of visible and/or palpable webs of string-like tissue extending subcutaneously from the axilla to the ipsilateral upper limb and leading to postoperative pain and limited shoulder range of motion (ROM) [6].

AWS represents a common postsurgical complication in BC survivors, its incidence varies between 6 and 91% in different studies depending on several risk factors [6,7]. More in details, invasive and extensive axillary surgeries, as axillary lymph node dissection (ALND), younger age at diagnosis and a body mass index ≤25 are known risk factors for AWS [8,9,10,11].

This detrimental condition usually develops within 5–8 postoperative weeks, affecting the subcutaneous tissue localized in one or more of the following sites: breast, axilla, medial arm, antecubital space, forearm, hand or lateral chest wall [12,13].

The etiology is not yet fully understood, although recent studies have shown AWS to be associated with damaged lymphatic vessels and lymphatic stasis after surgery associated with increased levels of local inflammatory mediators [14].

Even though AWS showed a spontaneous resolution in a period of three to six months in the studies conducted by Moskovitz et al. and Leidenius et al., we must consider that patients affected by this condition experience significant pain and reduced upper limb range of motion during these months. In addition, the presence of AWS could prevent patients from undergoing adjuvant radiotherapy when indicated because of the limited arm mobility [6,8,15].

Eventually, AWS is linked to secondary lymphoedema; patients who reported cording after BC surgery are at higher risk of developing breast cancer-related lymphoedema (BCRL), as described by Brunelle et al. More in detail, patients presenting with cording had 2.4 times the odds of developing BCRL compared to those who did not have cording [16,17].

Currently, there is neither standardized treatment nor national and international rehabilitative guidelines for the management of this complex clinical issue. Based on the literature, AWS-specific rehabilitation should last 4–5 weeks with 2–3 sessions per week with an average duration of 30–40 min for each session [17].

In a recent systematic review, Lippi et al. showed that the rehabilitative treatments proposed for AWS varied widely across different studies and were proposed in different combinations. Specifically, they found different manual therapy techniques to be effective. Among these techniques figure: manual lymphatic drainage (MLD); myofascial release techniques; cord, soft tissue and scar manipulation. Exercise therapies were also broadly represented in the studies they retrieved, including stretching exercises, resistance training and mobilization training. Eventually, the other therapeutic options that they found were Kinesio taping, compression bandages and intermittent pneumatic compression, compression garments, aqua lymphatic therapy and moist heat therapy [15,18].

Finally, the combination of a manual approach with therapeutic exercises could be an excellent management strategy for functional recovery and pain reduction in AWS patients, according to Agostini et al. [17].

Despite all these findings, there is a lack of knowledge about what should be the best rehabilitative approach for such patients. Thus, we sought to investigate whether our specific rehabilitative treatment protocol was safe, well-tolerated by the patients, and, eventually, if it was effective in resolving subcutaneous cording, reducing pain and improving arm mobility in patients affected by AWS.

## 2. Materials and Methods

We conducted an observational retrospective study, collecting data from medical records of BC survivors referred to the Oncological Rehabilitation Outpatients Service of an Italian University Hospital from January 2021 to June 2022.

The inclusion criteria were age ≥ 18 years, previous BC surgical treatment, and current diagnosis of AWS. We considered the following as exclusion criteria: breast cancer-related lymphoedema, concurrent malignant tumors, incomplete wound healing, acute vascular disease (i.e., thrombophlebitis), skin problems (i.e., infections), musculoskeletal comorbidities preventing patients from performing the rehabilitative treatment or representing possible confounders as pectoral muscle tightness, rotator cuff disease, adhesive capsulitis, low back pain, osteoarthritis, rheumatoid arthritis and ankylosing spondylitis.

Medical records have been consulted by physical medicine and rehabilitation physicians in order to collect reported data on the clinical evaluations carried out before (T0) and after (T1) the routinely administered AWS rehabilitation treatment offered by our center. T0 indicates the first rehabilitative clinical evaluation, during which diagnosis of AWS was formulated and the specific rehabilitation treatment protocol was proposed to the patient, while T1 refers to the medical examination carried out at the end of the treatment, when the resolution of the clinical issue was reported.

The AWS-specific treatment consisted of manual therapy (MT), stretching and mobilization of the affected upper limb and manual lymphatic drainage (MLD) sessions lasting 60 min. More in detail, MT consisted of soft tissue, scar and adherence mobilization techniques in association with stretching for tight subcutaneous cording: with patient lying on his back and the arm placed in available abduction, the mobilization technique performed consisted of transverse finger pressures applied perpendicular to the adherence axis; successively, the therapist performed a longitudinal tissue stretch along the adherence axis, between both thumbs or between the thumb and index, in order to strain the tight subcutaneous cording. In addition, upper limb stretching and passive mobilization involved shoulder abduction and flexion, elbow extension and wrist supination and extension movements. Finally, MLD was performed according to the most recent literature and clinical practice guidelines update of the Leduc’s method.

Specifically, treatment intensity, direction and depth were based on the clinical assessment of tissue glide and tightness revealed during the clinical evaluation. This approach has the advantage of offering a patient-tailored treatment rather than proposing the same treatment to each patient.

The sessions were held 3 times/week by specialized physiotherapists, up to the resolution of the clinical picture or interruption due to inter-current events.

The following data were retrieved from medical records:Demographic data: (a) sex, (b) age and (c) BMI.BC clinical and surgery data: (a) BC laterality, (b) BC histological subtype, (c) type of surgery (i.e., lumpectomy, quadrantectomy and mastectomy), (d) axillary lymph node dissection (ALND), (e) sentinel lymph node biopsy (SLNB) and (f) period of time elapsed between surgery and T0.Clinical data: (a) treatment adherence (i.e., treatment interruptions), (b) adverse events, (c) number of rehabilitation treatment sessions carried out, (d) AWS presence, (e) referred pain through the numeric rating scale (NRS) and (f) shoulder flexion and/or abduction range of motions (ROM).

### Statistical Analysis

Statistical analyses have been performed using Python 3.8 software, specifically Pandas, Numpy and Scipy packages.

Categorical variables were expressed as absolute numbers and percentages, whereas continuous ones were expressed as means ± standard deviations.

The one-sided binomial test was used to determine the significance of the AWS clinical resolution proportion at T1. The non-parametric Wilcoxon signed-rank test was instead applied to compare pre-intervention and post-intervention shoulder ROM and referred pain (NRS).

A *p*-value < 0.05 was considered statistically significant.

## 3. Results

### 3.1. Participants Characteristics

A total of 168 patients affected by AWS were first screened for eligibility. Only 92 of them met our inclusion criteria and were included in this study.

One patient stopped the rehabilitation treatment early due to the initiation of radiation therapy, and there were no adverse events, as shown in Figure 1.

Among the enrolled patients, 87 (95%) were woman. The mean age of our sample was 48 ± 4.24 years, and the mean BMI was 23.54 ± 3.54 kg/m^2^.

BC laterality was left in 46 (50%) patients, histological subtype was infiltrating ductal carcinoma in 76 (83%) patients, infiltrating lobular carcinoma in 12 (13%) of them and in situ ductal carcinoma in 4 (5%) patients.

The mean time elapsed between BC surgery and the first clinical evaluation (T0) was 42 days (CI 95%, 14–124), with a median of 32 days. The type of surgery was quadrantectomy in 22 (24%) patients and mastectomy in 70 (76%). A total of 61 (66%) subjects underwent SLNB, while 31 (34%) ALND surgery.

The mean number of rehabilitative sessions carried out by the remaining 91 patients was 8.74 ± 2.12.

All baseline characteristics are reported in Table 1.

### 3.2. Clinical Data

AWS was no longer clinically detectable at medical inspection and palpation after the rehabilitative treatment protocol in 58 patients (64% at T1 vs. 0 at T0; *p*-value = 0).

Shoulder range of motion was complete, reaching 180° of flexion and abduction, in a significantly higher proportion of patients at T1 (98% at T1 vs. 30% at T0; *p*-value = 1 × 10^−12^ and 98% at T1 vs. 29% at T0; *p*-value = 1 × 10^−12^, for flexion and abduction, respectively).

Finally, referred pain significantly decreased at T1 compared to T0 (*p*-value = 2.66 × 10^−11^). All clinical rehabilitation outcomes are shown in Table 2.

## 4. Discussion

Our observational study showed that a combination protocol composed of manual lymphatic drainage, manual therapies (including soft tissue, cording and scar mobilization), shoulder stretching and mobility exercises is a safe, well-tolerated and effective approach to manage AWS. Particularly, we had no adverse events while we obtained significant shoulder ROM improvement, clinical resolution of the subcutaneous cording and pain reduction in a large cohort of patients.

Similar results have been recently published by de Sire et al. in a case report. They proposed to a patient affected by a rare association between AWS and Mondor’s disease after breast surgery our same protocol based upon manual therapy (myofascial release techniques with soft-tissue mobilization, massage and manipulation of the tight cord and scar tissues), manual lymphatic drainage and therapeutic shoulder exercises, including stretching. The patient performed this 3 times/week protocol for 3 weeks and finally she obtained greater shoulder ROM, a disappearance of pain, an improvement in her quality of life and resolution of the cord-like indurations [19]. Considering their results, we decided to evaluate an analogous protocol on a larger sample, allowing stronger statistical inferences and evidence.

An association between MLD and physical therapy (PT) for the management of AWS and lymphedema was also studied by Cho et al. in 2015. They performed an RCT in a population of 48 women affected by post-surgical AWS, comparing 3 times/week for 4 weeks intervention consisting of MLD and physical therapy (PT) versus a control group treated with PT only. Physical therapy sessions included manual therapies (soft tissue mobilization and shoulder stretching and mobilization) in association with strengthening exercises and a 10-min warm-up and cool-down of stretching. Their purpose was to evaluate treatment effects on shoulder function, pain and lymphedema symptoms. They found that MLD and PT were significantly more effective than PT alone in reducing pain and arm volume. By contrast, quality of life and functional outcomes such as shoulder active ROM, shoulder flexor strength, DASH and the percentage of visible cords did not significantly differ between the two groups [20]. Interestingly, they proved MLD to be effective in pain reduction, but no significant impact on functional outcomes was found. Moreover, they focused on lymphoedema symptoms and arm circumferences. By contrast, we wanted to study MLD efficacy on AWS symptoms, excluding from our sample lymphoedema patients. This purpose was based upon previous studies supporting the role of manual lymphatic drainage and early stimulation of lymphatic flow in reducing local inflammatory mediators, which are thought to be part of the pathogenetic mechanisms underlying AWS, particularly supporting edema and pain [21,22].

Previously, in 2010, Moreau et al. enrolled 28 patients who developed AWS to study the differences between manual lymphatic drainage and adherence stretching versus upper extremity mobilization, soft tissue work and adherence stretching. This research article highlighted that both groups significantly reduced VAS and improved shoulder ROM, but no differences between groups were found. Thus, they concluded that both treatments played a positive role in recovering from AWS. Moreover, significant clinical improvement was detected in both groups after 10 and 13 treatments [23]. Curiously, we found a mean of 8.74 ± 2.12 rehabilitative sessions needed to obtain the clinical resolution. This value, which is slightly inferior to those reported by Moreau, could be explained by the higher intensity of our protocol composed of all the proposed treatments (MLD, stretching and mobilization and soft tissue techniques), consequently justifying the faster recovery from this syndrome.

It is crucial to underline the pivotal role played by tissue adherence of the cords, which represents the main limitation of this syndrome along with pain. Thus, manipulative treatment and soft tissue techniques, such as myofascial release, are broadly recognized as effective treatments for cord tissue limitations [17].

Specifically, regarding myofascial release, Ibrahim et al. conducted an RCT on sixty AWS women, comparing (1) direct myofascial release and kinesio tape, (2) direct myofascial release alone and (3) kinesio tape alone. Each group had a significant decrease in VAS and in cord thickness, but no differences between groups emerged [24]. Similarly, Lattanzi et al. published a case report on a patient affected by AWS extending distally down the upper extremity and proximally through the breast and trunk wall. She was treated with scar massage, soft tissue mobilization, myofascial release techniques, skin traction techniques, home stretches and a two-person stretch release technique. As a result, the cording significantly reduced and upper extremity function (valued through the DASH), shoulder ROM and muscle strength remarkably improved [25]. In addition, Jacob et al. presented the case of an AWS patient treated 7 months after BC surgery. The treatment included scar tissue techniques, cord stretching, self-massage, supportive bras and manual lymphatic drainage associated with a home protocol (self-lymph massage, compression garments, stretching exercises and aqua lymphatic therapy). The protocol provided for one 60-min session/week, for 6 weeks. Their main findings were represented by the improvement in VAS during shoulder ROM and the disappearance of cording [26]. Finally, Fourie et al. conducted a case report on a 47-year-old woman treated with manual soft tissue techniques, gentle stretching and self-mobilization. After 11 sessions of 30–45 min each, she obtained significant improvement in active and passive ROM, in tissue movement and glide, with no visible or palpable cording [27]. All these findings support our willingness to study an AWS combination protocol including manual therapies such as soft tissue, scar and adherence mobilization techniques together with manual lymphatic drainage and stretching and mobilization exercises whose efficacy was previously discussed.

To the best of our knowledge, this is the first study which evaluates a treatment protocol for AWS including all these techniques taken together on a large sample.

However, our study has some limitations that should be considered. Firstly, a retrospective monocentric design that could hinder any robust conclusion about the results obtained. Secondly, patients’ recruitment could represent a sample bias because we enrolled patients already referring to our service rather than enrolling all BC survivors with a diagnosis of AWS. Thirdly, subcutaneous cording is hardly a standardized parameter, thus we just focused on its presence or absence at pre- and post-treatment clinical evaluations without further investigations about qualitative and quantitative AWS variations.

Nonetheless, the present study might be considered a starting point for future investigations and, hopefully, for a randomized controlled trial evaluating the innovative AWS treatment protocols’ efficacy.

## 5. Conclusions

Taken together, our findings suggest that our AWS rehabilitative combination protocol is both safe and well-tolerated by patients. Moreover, it was effective in significantly reducing pain, improving shoulder flexion and abduction, and, finally, decreasing the subcutaneous stiff tissue cording. Furthermore, we observed these clinical improvements after a mean of 8.7 rehabilitative sessions.

In conclusion, our AWS rehabilitation protocol could be considered a promising treatment opportunity for these patients.

## Figures and Tables

**Figure 1 cancers-15-00426-f001:**
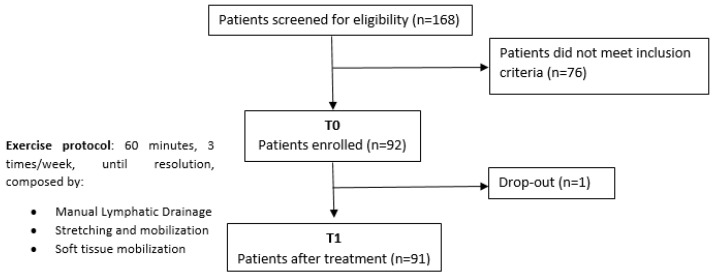
Study flowchart.

**Table 1 cancers-15-00426-t001:** Baseline demographic and clinical characteristics of patients.

Characteristics	No. (%) (Unless Otherwise Stated)
Sex	
Male	5 (5%)
Female	87 (95%)
Age (years), mean ± SD	48 ± 4.24
BMI, mean ± SD	24 ± 3.54
BC laterality	
Left	46 (50%)
Right	46 (50%)
BC subtype	
Infiltrating lobular	12 (13%)
Infiltrating ductal	76 (83%)
In situ ductal	4 (5%)
Time surgery-T0 (days), mean (CI 95%)	42 (14–124)
Type of surgery	
Lumpectomy	0
Quadrantectomy	22 (24%)
Mastectomy	70 (76%)
Axillary surgery	
SLNB	61 (66%)
ALND	31 (34%)
Treatment interruptions (patients)	1 (1%)
Rehabilitative sessions, mean ± SD	8.74 ± 2.12

SD: standard deviations; BMI: body mass index; BC: breast cancer; CI: confidence interval; SLNB: sentinel lymph node biopsy; ALND: axillary lymph node dissection.

**Table 2 cancers-15-00426-t002:** Clinical rehabilitative outcomes.

	T0 No. (%) (Unless Otherwise Stated)	T1 No. (%) (Unless Otherwise Stated)	*p*-Value
AWS absence	0	58 (64%)	0 *
Shoulder flexion complete	28 (30%)	89 (98%)	1 × 10^−12^ *
Shoulder abduction complete	27 (29%)	89 (98%)	1 × 10^−12^ *
Pain (NRS), mean ± SD	3.7 ± 2.27	1.77 ± 1.54	2.66 × 10^−11^ *

AWS: axillary web syndrome; NRS: numeric rating scale; SD: standard deviations; * statistical significance.

## Data Availability

The data presented in this study are available on request from the corresponding author.

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
