# Peer review of "Tolerability, Safety and Efficacy of a Specific Rehabilitation Treatment Protocol for Axillary Web Syndrome: An Observational Retrospective Study"

_cancers, 2023, doi:10.3390/cancers15020426_

Round 1

Reviewer 1 Report

This was an interesting report of a retrospective chart review for patients referred to a rehabilitation program for axillary web syndrome. It is clear, concise and well written.

It would benefit from a better description or appendix with the detailed protocol used for stretching, MLD, and massage so others could replicate it.

It will contribute to the literature on this under recognized problem.

Author Response

Dear Reviewer,

Thanks very much for you comment. We've added to the Materials and Methods section a detailed description of our treatment protocol, as you kindly suggested.

Reviewer 2 Report

The author describe an important and well done study of the management of axillary web syndrome in breast cancer patients. This can be a difficult management problem, and guidelines on diagnosis and treatment are very helpful. I have e few recommendations which may strengthen the manuscript.

The reference numbers are often omitted from the references. Please use a uniform system for references.

Line 97. Breast cancer related edema was an exclusion criteria. How of does edema, by standard criteria, accompany AWS?

Line 104. What is meant by T0 and T1? Please define.

Line 138. The observation period was define as January 2021 – June 2022. Line 138 indicates that 168 patients were affected with AWS, and 92 met inclusion criteria. These numbers would suggest that an extraordinary number of subjects had AWS in the 18 month period. Please clarify if this is correct.

Line 151. 66% of patients had SLNB, and 34% had AXLND. How many patients had both SLNB and AXLND.

Line 207: It is interesting that manual lymphatic therapy was applied in the absence of arm edema. Were these the standard compression maneuvers?

Conclusions. The authors provide an excellent description of the management of AWS. In retrospect, are there any additional components that should be included in future management of the syndrome. 

Author Response

Dear Reviewer,

Thanks very much for your comment. 

  • We've corrected the references numbers as you kindly siggested
  • Breast cancer related lymphoedema might present in people previously affected by AWS, a recent study reported that patients affected by AWS have 2.4 times the odds of developing BCRL compared to those who did not present AWS, making AWS a risk factor for developing BCRL. We've added these information in the main text (lines90-93). We've decided to use BCRL as an exclusion criteria because we wanted to assess safety, tolerability and efficacy of our treatment, which includes manual lymphatic drainage, on AWS only. BCRL could have be a confounder, especially for MLD. 
  • We've better defined T0 and T1 in the main text (lines 129-132). Thank you. 
  • Between January 2021 and June 2022 we've actually visited 168 patients with a formal diagnosis of AWS. We are a reference center for breast cancer rehabilitation, the surgeons of our Hospital's Breast Unit send us every patient for an evaluation 2-4 weeks after surgery. 
  • Every patient treated with ALND had previusly been treated with SLNB. Our protocol includes firstly the sentinel lymph node biopsy; successively, only if SLNB resulted positive, the patient underwent ALND. 
  • We included in our protocol the manual lymphatic drainge (MLD), in the absence of lymphedema, based upon previous study (lines 101) and based upon AWS etiology which relies on lymphatic vessels damages and lymphatic stasis after srgery (lines 82-83). This is our standard treatment.